# The Development of Digital Transformation and Relevant Competencies for Employees in the Context of the Impact of the COVID-19 Pandemic in Latvia

**Veronika Bikse [1], Inese Lusena-Ezera [1], Peteris Rivza [2] and Baiba Rivza [3,*]**

[1] Institute of Management Science, Liepaja University, 14 Liela Street, LV-3401 Liepaja, Latvia; vbikse@lu.lv (V.B.); inese.lusena-ezera@liepu.lv (I.L.-E.)

[2] Department of Computer Systems, Latvia University of Life Sciences and Technologies, 2 Liela Street, LV-3001 Jelgava, Latvia; peteris.rivza@llu.lv

[3] Institute of Economics and Regional Development, Latvia University of Life Sciences and Technologies, 2 Liela Street, LV-3001 Jelgava, Latvia

* Correspondence: baiba.rivza@llu.lv

**Abstract:** The current period describes the impact of the global COVID-19 pandemic and the ensuing economic crisis on businesses and the lives of citizens. It has accelerated digital transformation in all areas. The work and learning of many individuals have moved to the digital environment. In order to use digital technologies, employees need to acquire new knowledge and skills. The *aim* of this research study is to perform an analysis of the development of digital transformation and relevant competencies for employees and to identify the opportunities and challenges in Latvia. The research methodology applied for this research study is based on examining relevant theoretical concepts and publications of the EU regarding digital transformation. A survey method was used to find out the opinions of Latvian employers regarding the importance of digital transformation and relevant competencies for employees. The analysis of the research indicated that the majority of the respondents surveyed rated the level of implementation of digital transformation as high or medium-high, which shows that this is a good trend, and the digitalization process continues to progress. However, about a third of enterprises are only at the early stage of digitalization, while some have not yet begun it. The problem is the development of human capital competencies and digital skills. This is a specific research study that expands and provides insights into the situation in Latvia on the possibilities of implementation of digital transformation, which is closely linked with the development of human capital competencies and digital skills. This requires maintaining a holistic approach to targeted digital transformation management.

**Keywords:** COVID-19; competences; digitization; digitalization; digital transformation; sustainability

## 1. Introduction

Today, the greatest challenge is the COVID-19 pandemic and the resultant global economic crisis that has significantly changed people's habits of communication by means of information and communication technologies (ICT). The application of ICT has become the norm everywhere: at work, while learning, and for everyday transactions. Moreover, in particular, it has far-reaching implications for workers as it most likely has accelerated the transformation process of jobs [1]. Already, many organizations have successfully transformed their workflows. A very rapid transition to teleworking has occurred. Online work is becoming increasingly popular, and the number of employees working remotely tends to increase. In Latvia for example, only 4.8% of employees worked remotely before the COVID-19 pandemic crisis, while in 2020, it was already 39%, i.e., almost eight times more than in 2019 [2]. This is also evidenced by the results of surveys conducted by the Latvian Information and Communication Technology Association, which showed that 43% of enterprises provided opportunities for their employees to work remotely, and 45%

of enterprises used the possibility to receive and send e-invoices on a daily basis, while only a quarter used e-signatures (26%) [3]. Consequently, the coronavirus pandemic has forced enterprises to adopt digital transformation and change how they create, deliver, and capture value to their customers [4]. Moreover, in the near future, enterprises will face the additional challenge of big data management. If they do not master it, their competitive position will seriously weaken if it becomes the starting point of a genuine industrial revolution based on converging technologies [5].

In this context, more action is needed to implement the digital transformation in all areas. At the same time, to use the digital technology in different situations and for different purposes, employees will require and demand to acquire relevant competencies: new knowledge and skills, work organization and management skills, and other characteristics that become an important component of the development and competitiveness of individuals and enterprises. With digital transformation processes deepening, the most important issue is an appropriate, flexible education system to enable the development of competences and new skills. The mentioned aspects are closely interrelated: the better a company is technologically equipped and the more appropriate competencies and skills to use technologies employees have, the greater the opportunity to increase the competitiveness of enterprises and for the country to gain economic, social, environmental, and consumer benefits. However, it should be noted that a large part of the Latvian and EU population, 58% of EU individuals and 41% of Latvia's individuals, still lack basic digital skills and continue to build them up slowly [3]. In this context, an important research problem is that the digital transformation processes are not closely linked with the development of human capital competencies.

Therefore, the aim of the research is to perform an analysis of the development of digital transformation and relevant competencies for employees and to identify the opportunities and challenges in Latvia. However, digital transformation plays a critical role in an organization's ability to create new growth opportunities for businesses and improve the lives of citizens.

The questions addressed in this paper are as follows:

- What does digital transformation mean and what competences have to be developed to prepare new professionals?
- Are the digital transformation and the development of relevant competences in employees successfully carried out in all sectors of the national economy in Latvia?

## 2. Research Methodology

In order to achieve the aim set, the present research performed a review of scientific literature, examining publications and documents of the European Commission regarding the digital transformation. To obtain a deeper insight into the research problem, the implementation of digital transformation in the EU and Latvia is analyzed in this research. A survey method was used to find out the opinions of Latvian employers regarding the importance of digital transformation and relevant competencies for employees.

The survey was conducted on the Internet by filling out a specially designed online questionnaire at docs.google.com/forms. The questionnaire consisted of 10 questions. The structure of the questionnaire comprised open and closed questions. Several questions were formed based on the Likert Scale, allowing the respondents to provide their answers to several statements. A total of 162 respondents from Latvia participated in the survey. Replies were received from 161 employers or 99.4% of the total. In the survey (n = 161), the distribution of the employers by field of economic activity was as follows: 26.1% from the goods production sector, 14.3% from the services sector, 5.6% from the public sector, and 54.0% employers represented other fields, mostly through small and medium-sized enterprises.

According to the Latvian Statistical Database in the percentage distribution of economically active enterprises, in 2019, the production sectors (30.5%), business services (18.3%), public sectors (7.2%), and others, mostly small and medium-sized enterprises such as trade

and accommodation, real estate activities, as well as agriculture (54.0%) of all businesses in Latvia, were represented. Thus, all the questionnaires were considered suitable for the purposes of this survey. Most of the employers were relatively young people: 27.3% were under 30 years of age, 27.3% of the total were 31–50 years old, and 45.4% were over 50 years and older. The distribution of the employers by field of economic activity and age is shown in Figure 1.

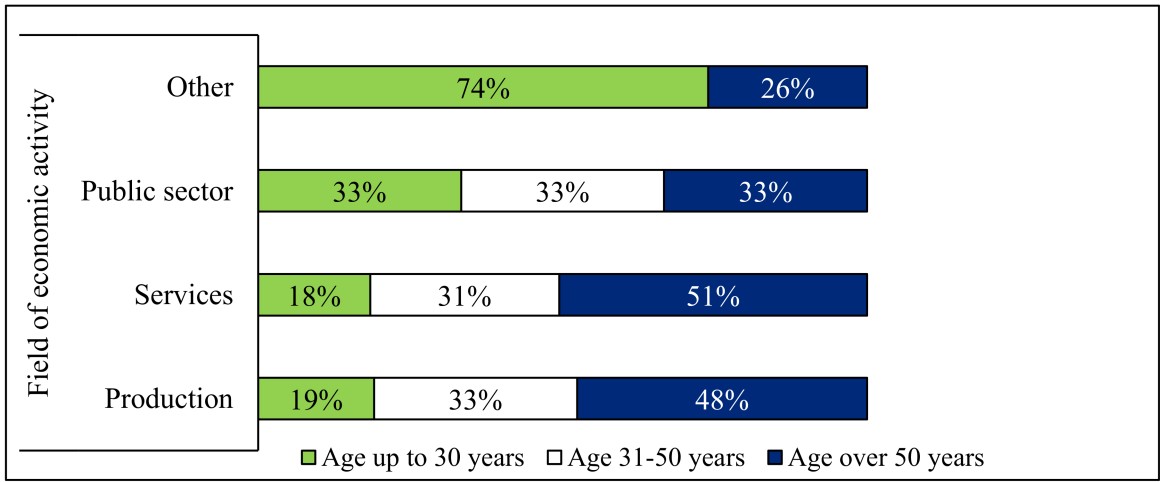

**Figure 1.** Distribution of the respondents by field of economic activity and age (n = 161).

The survey was carried out during the period between January and May 2020. The number of respondents was selected using a simple random sample. The survey results were analyzed and the data processed by the authors of the paper within the present study by applying methods of descriptive statistics (frequencies, central tendency, and crosstabs analysis), data visualization methods, and a nonparametric method—the Kruskal Wallis H-Test. SPSS software (26 version) and MS Excel 2016 were used to analyze the statistical data.

## 3. Results

### 3.1. What Does Digital Transformation Mean and What Competences Have to Be Developed to Prepare New Professionals?

The concept of digital transformation is used together with other concepts—digitization and digitalization. In this context, we should look at the relation between these concepts and the competences that have to be developed to prepare new professionals according to the challenges of the 21st century. This leads us to analyze the theoretical background of these concepts, which would then form the basis of the practical solution of the research problem. In this section, we present a general overview of these concepts.

#### 3.1.1. Digitization and Digitalization

Digital transformation goes beyond digitization and digitalization by including the whole organization. The lowest stage of digital transformation is digitization. A number of different definitions reviewed (studied) revealed that the concept of digitization is associated with the process of changing data into a digital form that can be easily read and processed by a computer [6]. According to Gartner's IT Glossary, digitization takes an analog process and changes it to a digital form without any different-in-kind changes to the process itself [7]. A similar description of the concept of digitization with small modifications mainly focusing on the process of changing data into a digital form has been given by [8,9]. For example, Wieberneit [8] defines digitization as the process of converting analog information into a computer-readable format with the goal of improving existing processes.

However, other authors, for example, Bloomberg [10], consider that it is the information being digitized, not the processes, is where digitalization comes in. Digitization essentially refers to taking analog information and encoding it into zeroes and ones so that computers can store, process, and transmit such information. Accordingly, the authors of the paper also consider that the concept of digitization does not relate to processes, while digitalization refers to the process in which digital technologies and digitalized data are used to create new processes to focus on potential changes in the processes beyond the mere digitizing of existing processes and forms [11].

### 3.1.2. The Concept of Digital Transformation

In the last years in the specialized literature, policy-related papers, and reports, much attention has been paid to digital transformation. For example, Verina and Titko [12] summarized a number of publications devoted to digital transformation based on the findings of various authors' papers from the Web of Science and SCOPUS scientific databases from 1995 to 2018. The authors point out that there are plenty of definitions provided by academicians, government authorities, and business experts and more than 3000 relevant publications, the number of which is increasing every year.

The literature review shows that there are various approaches to defining the concept of digital transformation. Therefore, digital transformation has a definition problem [13]. Some authors define digital transformation by mainly focusing on the transformation of existing digital technologies to create new ones [14–17]. For instance, Duncin [15] defines the digital transformation as the process of using digital technologies to create new—or modify existing—business processes, culture, and customer experiences to meet changing business and market requirements.

In a broader sense, digital transformation is presented as the change of organizational processes. Wieberneit [8] indicated in their research that a digital transformation is an organizational transformation that covers organization, values, culture, mission, and vision, using an outside-in view. It is enabled by computer technology.

Most other authors [9,11] consider that digital transformation is more relevant to individuals, not digital technology. For example, Talin [9] pointed out that it is important to understand that digital transformation is never triggered by technology; it is always about solving a problem or providing a new approach to customers. The customer-centric solution is always the start of the digital transformation, not the technology. (Do not create problems by looking first at tech and then only focusing on a solution for this technology).

In addition, other authors [12,18,19] have analytically summarized essential definitions from different sources and concluded that the given definitions allow categorizing the digital transformation (DT) into three distinct elements: technological, where DT is based on the use of new digital technologies such as social media, mobile, analytics, or embedded devices; organizational, where DT requires a change of organizational processes or the creation of new business models; and social, where DT is influencing all aspects of human life. In this regard in their research, Verina and Titko [12] pointed out that digital transformation is not about the implementation of IT solutions only. However, it should be viewed in a broader context as "organizational change", "cultural transformation", and "moving toward a customer-centric approach". According to Bloomberg [10], "digitization and digitalization are essentially about technology, but the digital transformation is not. Digital transformation is about the customer". In this regard, the element "people" becomes essential and even more important than anything else. Therefore, we can conclude that the concept of digital transformation is more comprehensive than the concepts of digitization and digitalization. In order to ensure digital transformation, it requires maintaining a holistic approach to ensuring the implementation of all the above-mentioned elements, digital technologies and organizational solutions together with the human element. According to the European Commission, [20] "machines and humans can do together". Moreover, the increasing use of information technologies requires addressing the development of human capital: adequate knowledge, skills, and specific competences, without which one

cannot fully benefit from digital transformation. It makes us seek answers to the questions of how to explain what competences and personal traits must be developed for digital transformation and prepare new professionals according to the challenges of the labor market.

### 3.1.3. Human Capital Competencies and Digital Skills

Based on the academic literature and the review of policy documents in the field of developing relevant skills and competences for digital transformation, one can find that some authors mainly focused on developing digital skills. Others emphasized the importance of digital competences and related skills in the implementation of digital transformation [21–24]. The importance of digital competencies/skills and their development is well-recognized in most European Commission policy documents. For example, the Digital Education Action Plan states that Member States should pay special attention to increasing and improving the level of digital competences across all segments of the population [25].

It is evident that for digital transformation, it is important for everyone to build up relevant competencies and digital skills that are required when using ICT and digital media because, first, during digital transformation, to be able to deal and work more effectively with the latest available technologies, one should have a high level of professionalism with advanced digital skills. This was also evidenced by the survey of residents of Latvia conducted by SKDS in 2020, in which more than 1000 respondents aged 18–75 participated. Most or 72% of respondents believed that they needed to improve their digital skills. Additionally, 80% of respondents had done something to improve their digital skills in the last year [26].

Second, it is important to develop not only skills but also digital competences, as the concept of digital competence is much broader than the concept of digital skills. Digital competence is a combination of knowledge, skills, and attitudes, including the development of soft skills such as problem solving, collaboration, and creativity. Consequently, digital skills are an important component of digital competence. Moreover, OECD [27] documents stress that the concept of digital competence must be focused on a broader approach, as in the digital economy era, ICT skills will not be enough, and other complementary skills will be needed. as Additionally, the European Commission policy documents states that "in addition to digital skills, the digital economy requires also complementary skills such as adaptability, communication and collaboration skills, problem solving, critical thinking, creativity, entrepreneurship and readiness to learn" [25] (p.13).

However, it should also be noted that during the COVID-19 pandemic, individuals are in a completely different environment, mainly working, doing daily activities, and learning remotely. Everyone who must deal with remote activities becomes like a "self-manager" person. S/he should be ready to work independently, plan his/her working time, make optimal decisions, organize his/her work, continuously acquire new knowledge, and tackle other problems and must be a high-qualified specialist in a certain field. Moreover, each individual person has to become the subject of social changes and must be able to understand the complicated processes of societal development and influence them. This means that the role of individuals in the production process, learning, studies, and daily life is significantly modified, and the focus is placed on a competent individual as the most important precondition for competitiveness. Therefore, the European Commission developed a framework of eight key competences as part of their lifelong learning strategies that have to be developed by everyone, beginning from childhood and throughout their entire life. These competences are considered equally important for life and work in a knowledge-based society [28].

Accordingly, the authors of the paper consider that the process of digital transformation will require developing the individual's key competences. It envisages implementing competence-based education together with digital education, thus integrating the elements of digital education in all study programs (courses) focused on building key competences. This will encompass a range of skills, from technical, academic, sectoral, and digital skills to

softer skills like problem solving, creative and design thinking, communication, emotional intelligence, multicultural openness, leadership, and managerial and interaction skills [29]. It means that the digital economy will require all people to build up a range of new knowledge, abilities, and different digital skills needed not only in traditional STEM occupations and ICT professionals but also in nearly all job sectors where ICT complements existing tasks [30].

*3.2. Are the Digital Transformation and the Development of Relevant Competences in Employees Successfully Carried out in All Sectors of the National Economy in Latvia?*

To answer these questions, an analysis of the implementation of digital transformation in the EU and Latvia, the employers' ratings of the importance of digital transformation, the relevant competencies for employees in their organizations, and the challenges they were facing was carried out.

3.2.1. Implementation of Digital Transformation in the EU and Latvia

The importance of implementing the digital transformation in all areas is highlighted in multiple European Union (EU) strategy and policy documents. One of the main political priorities of the European Commission for 2019 to 2024 is to shape the EU so that it is fit for the digital age and empower its citizens with a new generation of technologies [31]. Furthermore, according to the European Commission forecasts for 2019 to 2030, one of the five key global broad trends in the EU is a revolution in technologies, and digitization transforms all aspects of society, such as politics, governance, education, science, lifestyles, collective intelligence networks, the setting-up of open systems, and health, including the transformation of the human genome. Divisions between education, work, leisure, and retirement phases will be less clear-cut than today, and training will be life-long for many [5].

An assessment of the current state of implementation of digital transformation in the EU and Latvia is based on the data from the database of the Digital Economy and Society Index (DESI, 2020). It provides a much-needed, integrated information source on Europe's overall digital performance, tracks the progress of EU countries in digital competitiveness, and is a solid decision-making basis for policy development [32].

The Digital Economy and Society Index (DESI, 2020) shows that across the EU Member States, there was impressive progress in digital transformation over the last 5 years. Finland, Sweden, Denmark, and the Netherlands have the most advanced digital economies and lead the ranking of all the EU Member States (Figure 2).

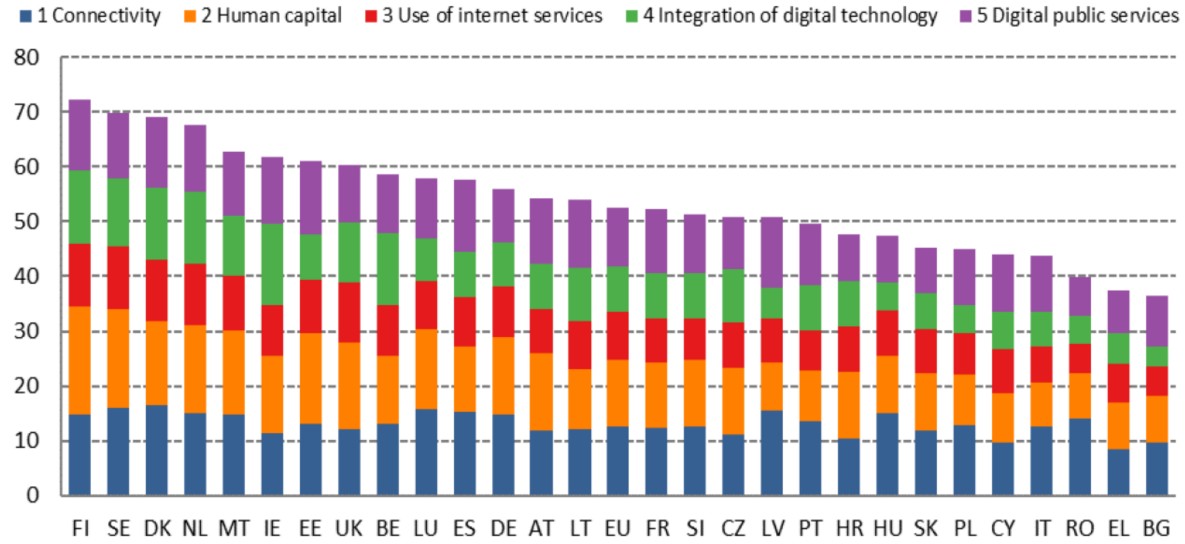

**Figure 2.** Rankings of EU Member States on the Digital Economy and Society Index in 2020 based on 2019 data [32].

Figure 2 shows that Latvia was ranked 18th among the 28 EU Member States. This position of Latvia could be influenced by the low values of indicators in the area of integration of digital technology. Bulgaria, Greece, Romania, and Italy had the lowest scores on the index.

It is important to underline that according to the rankings (DESI, 2020), large enterprises were becoming more and more digitized: 38.5% of them relied already on advanced cloud services and 32.7% were using big data analytics. The top EU performers in the digitization of businesses are Ireland, Finland, Belgium, and the Netherlands. The quality and usage of digital public services also increased: 67% of internet users who submitted forms to their public administration now use online channels (up from 57% in 2014). The top performers in this area were Estonia, Spain, Denmark, Finland, and Latvia. Throughout the past years, there has been an improvement in Internet user skills, with 58% of individuals having at least basic digital skills, 33% having above-basic digital skills, and 61% of individuals having at least basic software skills [32].

Similarly, in Latvia, significant progress has also been made in the digital transformation of some areas in recent years. For example, Latvia ranked second in the world in terms of mobile Internet usage. During the last three years, the use of the Latvian Mobile Phone (LMT) network has increased three times, and during the pandemic by another third [33]. Latvia is among the top EU Member States with the main public services reachable online for citizens and businesses. According to the Digital Economy and Society Index (DESI, 2020), Latvia ranked fifth in terms of e-government performance: e-signature, electronic documents, and digital mail. The introduction of automatically partially completed forms on the service portals of public institutions has been particularly successful, reaching an index value of 86 against the EU average of 59. In this field, Latvia had the highest score (ranked fourth).

The digital transformation is being successfully carried out in the banking and insurance sector, as well as in several large enterprises. Many enterprises have been able to respond quickly and reorient their business in the changing environment by digitizing the services provided and even creating and offering new, innovative products and solutions to the market. The results of a RAIT GROUP survey "Use of Telecommunication Services by Enterprises" conducted in May–July 2020 show that there was a significant increase in the use of various digital solutions by 10 percentage points compared with the previous year. The importance of websites has grown—without them, no company can do business— and the amount of data stored remotely increased, while the domain of the company has become 12% more important. The use of Office 365 in office and corporate work as well as record keeping has tripled, while accounting software in the form of cloud services has also been actively used by small and medium-sized enterprises (SMEs) [33].

Despite the progress, the most essential problem of digital transformation and developing relevant competences pertains to the small and medium-sized business segment because, during the COVID-19 pandemic, many Latvian SMEs were not ready for change and could not adapt quickly to the changing market conditions in order to modernize their production processes and promote the integration of digital technologies. For example, the European Investment Bank EIBIS survey that gathers information on investment activities among small and medium-sized enterprises (SMEs) and larger corporations showed that they expected to use digital technologies in the long term, and Latvia's SMEs had a lower score than the EU average (22% and 43%, respectively). As for the implementation of digital technologies across sectors, Latvia lagged behind the EU averages: firms in the manufacturing sector (18% versus 55%) and services and infrastructure sectors (18% versus 49%). Only firms in the construction sector had a relatively higher score compared with the EU average (55% versus 37%) [32,34–36].

In this context, according to the DESI 2020, Latvia had one of the lowest scores on the index of digitization of businesses and e-commerce compared with the EU average. The leading countries were Ireland, Finland, Belgium, the Netherlands, Denmark, and Sweden, with scores greater than 55 points (out of 100). At the other end of the scale,

Bulgaria, Romania, Hungary, Poland, Greece, and Latvia lagged behind with scores less than 35 points, significantly below the EU average of 43 points. Moreover, only 8% of Latvian enterprises used big data, 11% used cloud computing services, and 11% had web sales to customers [32].

In addition, developing human capital competencies and digital skills is a problem because no one can fully benefit from digital technologies. According to the *human capital dimension* of the DESI 2020, the level of digital skills has continued to grow slowly, and among Latvia's individuals, at least 41% had basic digital skills and 24% had above-basic digital skills. In terms of digital skills, Latvia ranked the lowest in both sub-dimensions of human capital, followed by Greece, Bulgaria, Romania, and Italy. However, specialists in the field of information and communication technologies made up only 1.7% of the total number of employees in Latvia. This was almost half of the EU average of 3.9% [32].

Consequently, it has to be mentioned that the situation in the EU was very diverse. Digital transformation and developing relevant competences for employees occurred in some countries successfully, whereas in others slowly. In addition, digital solutions for Latvian enterprises and organizations are very different. The solutions were of much lower quality in the private sector and small and medium-sized enterprises than those in larger enterprises and the public sector. Therefore, in the following sections, we will describe employers' ratings of the importance of the above-mentioned issues in their organizations and the challenges they were facing and the levers executives can use to drive digital transformation in Latvia.

### 3.2.2. Employers' Ratings of the Importance of Digital Transformation

As part of the survey, the employers were asked to rate the implementation of digital transformation in their organizations on a scale from 0 to 10 (0—no digitalization in their enterprises at all; 10—the highest). The respondents' ratings are summarized in Figure 3.

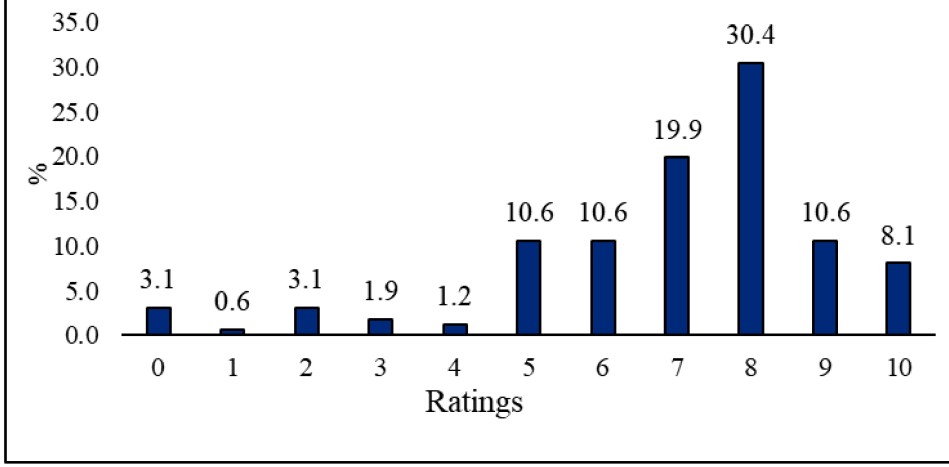

**Figure 3.** Percentage breakdown of the respondents' ratings of implementation of digitalization in their organizations (n = 161).

As shown in Figure 3, the majority of the respondents surveyed rated the implementation of digital transformation in their organizations as high or medium-high in the range from 7 to 10. However, almost a third rated it as relatively low. Only 3.1% indicated that there was no digitalization in their enterprises at all; however, a total of 5.6% of employers rated the implementation as unsuccessful (scale from 1–3). This could relate to SMEs, which were newly established, had relatively young leaders, and were at the early stage of digitalization.

The mean rating of implementation of digitalization by the employers was 6.9 (SD = 2.2), while the most frequent rating given by the employers was 8 (Table 1). As shown by the central tendency indicators of the ratings of implementation of digitalization summarized

in Table 1 by field of economic activity and employer age, it is not observed that in any of the fields of economic activities and employers' age groups, the average rating would be significantly higher or lower. Comparing the rating results using a Kruskal Wallis test, no statistically significant difference in the ratings of implementation of digitalization was found between the sectors of goods and services, the public sector and other fields ($p = 0.149$), and the age groups of employers ($p = 0.939$) (Table 2).

**Table 1.** Ratings of implementation of digitalization by field of economic activity and employer's age: statistics of central tendency.

| | | Implementation of Digitalization in Organization | | | | | |
| --- | --- | --- | --- | --- | --- | --- | --- |
| | | Mean | Median | Mode | Max | Min | Standard Deviation |
| **Field of Economic Activity** | Production | 6.5 | 7.0 | 8.0 | 10.0 | 0 | 2.5 |
| | Services | 7.2 | 7.0 | 8.0 | 10.0 | 1.0 | 1.9 |
| | Public sector | 7.7 | 8.0 | 8.0 | 10,0 | 2.0 | 2,3 |
| | Other | 6.5 | 7.0 | 8.0 | 10.0 | 0 | 2.6 |
| | Total | 6.9 | 7.0 | 8.0 | 10.0 | 0 | 2.2 |
| **Employers Age** | Up to 30 years | 6.8 | 7.5 | 8.0 | 10.0 | 0 | 2.7 |
| | 31–50 years | 6.7 | 7.0 | 8.0 | 10.0 | 0 | 2.4 |
| | Over 50 years | 7.1 | 7.0 | 8.0 | 10.0 | 2.0 | 1.8 |
| | Total | 6.9 | 7.0 | 8.0 | 10.0 | 0 | 2.2 |

**Table 2.** Implementation of digitalization in organization by field of economic activity: Kruskal Wallis test results.

| Test Statistics [a] | | |
| --- | --- | --- |
| | Field of Economic Activity | Employer's Age |
| Kruskal Wallis H | 5.326 | 0.126 |
| df | 3 | 2 |
| Asymp. Sig. | 0.149 | 0.939 |

[a.] Kruskal Wallis test.

Recent studies on the digital transformation potential of SMEs (150 SMEs surveyed) also showed that most SMEs in Latvia are at the first stage of digital transformation (digitization) [36]. SMEs still devote 64% of their activities to the provision of analog information, calling and sending letters to customers, as well as the circulation of paper documents. However, an assessment of the use of technologies by SMEs reveals that the degree of digitalization is 38%. The most popular technologies are cloud computing and social media with a utilization rate of 66%. The least popular was robotics with 21% [36]. The results of the present research are considerably consistent with the DESI 2020 and the European Investment Bank EIBIS survey, which showed that the implementation of digital technologies across sectors in Latvia lagged behind EU averages and had one of the lowest scores on the index of digitization of businesses and e-commerce [32,34–36].

Even though the ratings of implementation of digitalization did not differ significantly across the age group of employers, the analysis of how the employers of different ages rated the level of digitalization in their organizations shows that the employers in the age group over 50 gave much higher ratings (Figure 4). To our knowledge, this could be because a significant number of employers represented areas where the digital transformation was already being carried out successfully, such as the banking and insurance sector, as well as several large enterprises.

To identify to what extent digitalization was developed in the organizations represented by the employers, the questionnaires sent to them gave them the opportunity to rate the different levels of digitalization processes implemented. Each of the levels mentioned in the questionnaire was rated by the employers based on the number of digital processes implemented as follows: 0—no digitalization was implemented; one to three processes

were digitalized; three to five processes were digitalized; and more than five processes were digitalized. The employers' ratings are summarized in Table 3. An analysis of the employers' total ratings of the level of digital transformation in their organizations shows that despite the fact that about a third of enterprises (32.9%) were only at the early stage of digitalization, implementing one to three processes, the majority of them (63.3%) had digitalized from three processes to more than five processes. It was found that the level of digital transformation differed between the sectors of economic activity ($p = 0.026$) (Table 4). The results of the comparative analysis in Table 3 show that the digitalization of more than five processes has been implemented more by the public sector (66.7%), while in the goods sector, it has been implemented by 21.4% of enterprises, and in the sector of services by 32.2% of enterprises.

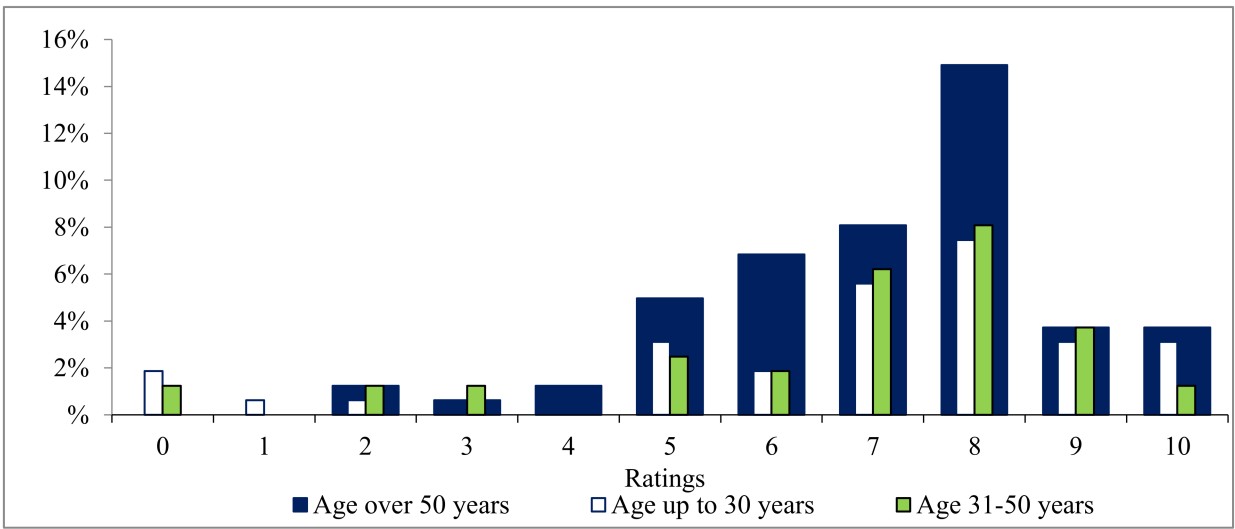

**Figure 4.** Percentage breakdown of the employers' ratings of implementation of digital transformation in their organizations by age. Calculated from the total percentage of the ratings (n = 161).

**Table 3.** Percentage breakdown of the employers' ratings of the level of digital transformation in their organizations by field of economic activity (n = 161).

|  |  | Field of Economic Activity | | | | |
| --- | --- | --- | --- | --- | --- | --- |
|  |  | Production | Services | Public Sector | Other | Total |
|  | No digitalization was implemented | 7.1% | 2.3% | 0% | 4.3% | 3.7% |
| **Digital** | 1–3 processes were digitalized | 38.1% | 29.9% | 11.1% | 43.5% | 32.9% |
| **transformation in** | 3–5 processes were digitalized | 33.3% | 35.6% | 22.2% | 30.4% | 33.5% |
| **organizations** | More than five processes were digitalized | 21.4% | 32.2% | 66.7% | 21.7% | 29.8% |
|  | Total | 100.0% | 100.0% | 100.0% | 100.0% | 100.0% |

**Table 4.** Ratings of the level of digital transformation in organizations by field of economic activity: Kruskal Wallis test results (n = 161).

| Test Statistics [a,b] |  |
| --- | --- |
|  | **Digital Transformation in Organizations** |
| Kruskal Wallis H | 9.237 |
| df | 3 |
| Asymp. Sig. | 0.026 |

[a.] Kruskal Wallis test. [b.] grouping variable: field of economic activity.

As it turned out, three to five processes and more than five processes were digitalized in the enterprises. In this respect, the employers were asked to assess whether their

employees and enterprises were ready for digital transformation. A data analysis of the survey indicates that the employees' readiness for digitalization had a mean rating of 6.6 (SD = 2.5) and the enterprises' readiness for digitalization had a mean rating of 6.4 (SD = 2.8), which means that the employers rated the readiness from medium to high (on a scale from 0 to 10) (Table 5). Comparing the results between the sectors of economic activity and between the age groups of employers, it was revealed that there were statistically significant differences in the ratings of employees' readiness ($p = 0.002$) and enterprises' readiness for digitalization ($p = 0.010$) between the sectors of economic activity (Table 6). The results of the comparative analysis summarized in Table 5 show that among the sectors of economic activity, the lowest mean rating was given by the employers from the goods sector both for the readiness of their employees (5.3 (SD = 3)) and the readiness of the enterprises (5.4 (SD = 3.2)) themselves for digitalization. An analysis of the results shows that the average rating of the public sector for the readiness of the employees was 7.7 (SD = 2) and 8 (SD = 2.1) for the readiness of the enterprises (Table 5).

**Table 5.** Employers' ratings of their employees' and enterprises' readiness for digital transformation by employer's age and field of economic activity: central tendency (n = 161).

| | | Up to 30 Years | Age 31–50 Years | Over 50 Years | Production | Services | Public Sector | Other | Total |
|---|---|---|---|---|---|---|---|---|---|
| | | | | | | | **Field of Economic Activity** | | |
| Employees' readiness for digitalization | Mean | 6.7 | 6.7 | 6.4 | 5.3 | 7.1 | 7.7 | 6.3 | 6.6 |
| | Median | 7.0 | 7.0 | 7.0 | 6.0 | 7.0 | 8.0 | 7.0 | 7.0 |
| | Mode | 7.0 | 7.0 | 7.0 | 7.0 | 7.0 | 8.0 | 7.0 | 7.0 |
| | Max | 10.0 | 10.0 | 10.0 | 100 | 10.0 | 10.0 | 10.0 | 10.0 |
| | Min | 0 | 0 | 0 | 0 | 0 | 3.0 | 0 | 0 |
| | Standard Deviation | 2.7 | 2.3 | 2.5 | 3.0 | 2.0 | 2.0 | 3.0 | 2.5 |
| Enterprises' readiness for digitalization | Mean | 6.7 | 5.8 | 6.5 | 5.4 | 6.7 | 8.0 | 6.3 | 6.4 |
| | Median | 7.5 | 7.0 | 7.0 | 6.0 | 7.0 | 8.0 | 8.0 | 7.0 |
| | Mode | 7.0 | 70 | 7.0 | 7.0 | 7.0 | 8.0 | 8.0 | 7.0 |
| | Max | 10.0 | 10.0 | 10.0 | 10.0 | 10.0 | 10.0 | 10.0 | 10.0 |
| | Min | 0 | 0 | 0 | 0 | 0 | 3.0 | 0 | 0 |
| | Standard Deviation | 3.2 | 3.0 | 2.5 | 3.2 | 2.5 | 2.1 | 3.4 | 2.8 |

**Table 6.** Employers' ratings of their employees' and enterprises' readiness for digital transformation by employers age and field of economic activity: Kruskal Wallis test results (n = 161).

| | **Test Statistics** [a] | | | |
| | **Age** | | **Field of Economic Activity** | |
|---|---|---|---|---|
| | Employees' readiness for digitalization | Enterprises' readiness for digitalization | Employees' readiness for digitalization | Enterprises' readiness for digitalization |
| Kruskal Wallis H | 1.106 | 4.204 | 14.569 | 1.328 |
| df | 2 | 2 | 3 | 3 |
| Asymp. Sig. | 0.575 | 0.122 | 0.002 | 0.010 |

[a.] Kruskal Wallis test.

Based on the analysis of the surveys of employers' ratings of the importance of digital transformation, we can conclude that the majority of the employers surveyed rated the implementation of digital transformation as medium-high and that from three to five processes and more than five processes were digitalized in their organizations as well as that their employees and enterprises were ready for digital transformation in general. This is a positive trend, which shows that digitalization processes continue to develop, the majority of organizations are ready for changes in the field of digitalization, and one can hope for positive changes in the future. However, about a third of enterprises are only at the early stage of digitalization, while some have not yet begun implementing it.

3.2.3. Employers' Ratings of the Importance of Relevant Competencies for Employees

As mentioned above, the digital transformation and the impact of the COVID-19 pandemic on businesses and on the lives of citizens significantly modifies the role of

individuals. Under these conditions, the development of human capital might be explained in a much broader sense as the development of the individual's key competences, including digital competence/skills. It is built through a process of acquiring knowledge, skills, and experience.

To our knowledge, the development of human capital could be viewed in two senses: in a broader sense, it is oriented toward developing a certain set of personal traits and soft skills, which are important in any field of activity and in life, without directly associating it with how to use ICT, while in a narrow sense, it involves developing specific skills and acquiring knowledge of and experience in how to use ICT. Within this context, the answers of the respondents to the open question about how they rate the importance of relevant competencies for employees associated with digital transformation are also grouped in Table 7. on the one hand as specific skills, while on the other hand as soft skills.

**Table 7.** Employers' ratings of the importance of relevant competencies for employees associated with digital transformation.

| | Comments |
|---|---|
| **Specific skills** | Ability to work with technology; knowledgeable, competent<br>Programmer with good communication skills<br>Ability to achieve better results with fewer resources<br>Loyal, trustworthy<br>Accuracy at work, honesty<br>Competent in the field<br>Ability to speak several languages; a good partner, competent in the industry<br>Ability to improve the e-environment |
| **Personal traits and soft skills** | Ability to work in a team<br>Innovative; ability to make decisions, ability to co-operate with others<br>Ability to listen and help with solving problems<br>Rich in initiative, creative, ready for change, ability to adapt to new conditions and technologies, focused on cooperation<br>Willingness to work, good communication and organizational skills<br>Knowledgeable and willing to develop and grow<br>Supportive, enthusiastic, rich in ideas<br>Flexible, self-motivated<br>Willing and able to work independently and effectively, motivated<br>Communicative with new ideas, with a desire to implement them, good communication skills<br>Wish to keep up with the times, learn and apply the acquired knowledge in practice<br>Ability to control emotions, act rationally in a stressful situation, take responsibility, think outside the box |

As shown in the table, the respondents indicated in their comments that, along with the development of specific digital skills, it is essential that individuals also have personal talents, traits, and abilities such as the ability to cooperate with others, work in a team and have tolerance, start new activities, show initiative and enthusiasm, put forward real aims and try to achieve them, and act creatively, as well as other personal qualities. Similar answers were given to the question of what the most important problems in their organization with regard to digital transformation are. The employers stressed the need to develop specific digital skills such as the ability to work with digital devices, be competent in the latest technologies, and work in difficult conditions, as well as soft skills. The employers' replies are summarized on a scale from 1 to 10 (1—the least important; 10—the most important) in Figure 5.

The analysis of the survey data revealed that these problems in organizations with regard to digital transformation did not differ significantly between the sectors of economic activity ($p > 0.05$) and between the ages of employers ($p > 0.05$) (Table 8).

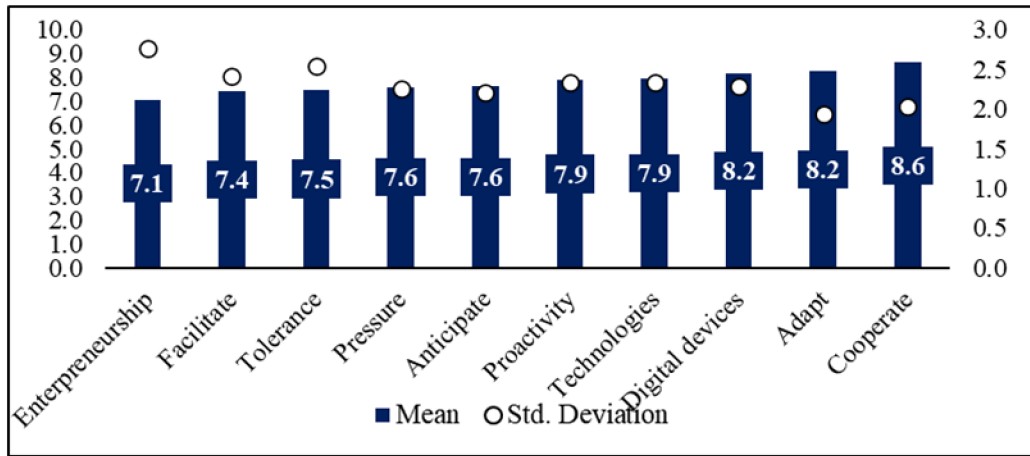

**Figure 5.** Employers' ratings (on a scale of 1 to 10) of the most important problems associated with digital transformation.

**Table 8.** Employers' ratings of the most important problems associated with digital transformation by field of economic activity and employers' age: Kruskal Wallis test results.

| | Test Statistics [a] | | | | | |
| | Field of Economic Activity | | | Age | | |
| | Kruskal-Wallis H | df | Asymp. Sig. | Kruskal-Wallis H | df | Asymp. Sig. |
|---|---|---|---|---|---|---|
| Digital devices | 4952 | 3 | 0.175 | 0.146 | 2 | 0.930 |
| Technologies | 0.734 | 3 | 0.865 | 0.555 | 2 | 0.758 |
| Pressure | 0.327 | 3 | 0.955 | 0.482 | 2 | 0.786 |
| Adapt | 3.215 | 3 | 0.360 | 0.048 | 2 | 0.976 |
| Cooperate | 3.476 | 3 | 0.324 | 1.866 | 2 | 0.393 |
| Facilitate | 3.015 | 3 | 0.389 | 1.292 | 2 | 0.524 |
| Anticipate | 5.183 | 3 | 0.159 | 5.114 | 2 | 0.078 |
| Proactivity | 0.584 | 3 | 0.900 | 0.917 | 2 | 0.632 |
| Entrepreneurship | 2.681 | 3 | 0.444 | 1.458 | 2 | 0.482 |
| Tolerance | 0.313 | 3 | 0.958 | 5.402 | 2 | 0.067 |

[a.] Kruskal Wallis test.

As shown in Figure 5, the highest rating of 8.6 (SD = 2.0) showed that the employers believed that the greatest challenge was to develop employees' abilities to communicate with others, as well as to work with digital devices and adapt (8.2, respectively) to digital transformation. The fact that the employers considered the development of employees' soft skills to be one of the most important problems, to our knowledge, related to the need for a faster transition to remote work. It does not require a permanent presence in the enterprise or institution. Job functions can also be performed from home via the Internet. Under the new circumstances, job duties are increasingly performed based on self-organization at the workplace, the ability to work independently and simultaneously coordinate one's actions with other partners of a united team, and to choose an optimal solution in multi-variant situations. This, in turn, requires the activation of abilities and skills such as planning, making decisions, communication, the desire to take up responsibility, independence, and the ability to see (find) a certain problem and solve it.

## 4. Discussion and Future Research Recommendations

The literature review shows that there are various approaches to defining the concept of digital transformation. Some authors define digital transformation by mainly focusing on the transformation of existing digital technologies to create new [14–17]. Other authors [9,11] consider that digital transformation is more relevant to individuals, not digital technology.

Our position on the main elements of digital transformation identified by various authors could be supported in general. At the same time, the authors consider that the elements of digital transformation they developed are more authentic and more practical [12,18,19]. It has been argued that the concept of digital transformation is more comprehensive than the concepts of digitization and digitalization. Digital transformation (DT) includes three main elements: technological, organizational, and social. In order to ensure digital transformation, it requires maintaining the holistic approach to ensuring the implementation of all the above-mentioned elements—digital technologies and organizational solutions together with the human element. The authors consider that a mentioned approach to the concept of digital transformation is essential for the practical solution of the research problem and is a great opportunity for sustainable economic development.

Today, individuals are in a completely different environment, mainly working, doing daily activities, and learning remotely. This means that the role of individuals in the production process, learning, studies, and daily life is significantly modified, and the focus is placed on a competent individual and the development of human capital: adequate knowledge, skills, and specific competences, without which one cannot fully benefit from digital transformation. Moreover, the authors of the paper consider that the process of digital transformation will require developing not only digital skills or digital competences [21,24] but also the individual's key competences. In order to develop key competences, personal specific skills, and qualities in an individual, it is necessary to provide the individual with an opportunity to acquire new knowledge and relevant competencies to be able to live and work in the new digital environment. In this respect, an analysis of the views of the respondents on the open question "What additional competencies do employees need to be ready for the digital age?" shows that it is important for their employees to be trained in the use of specific equipment and technologies, be able to understand and work with the latest technologies, acquire IT knowledge, and be ready for continuous learning and development, as well as learn several foreign languages, thus expanding their personal fields of competences. At the same time, according to a survey of residents of Latvia conducted by SKDS, businesspersons and entrepreneurs did not take sufficient care of the further education of their employees, as the majority or 62% of respondents had independently tried to acquire knowledge and improve their skills in working with technologies during the last year. Only 17% of respondents indicated that training opportunities were provided by their employers [33]. This is not a positive trend, and as mentioned earlier, the main problem in Latvia with regard to the implementation of digital transformation was a lack of qualified specialists and the fact that a large segment of the population lacks even basic digital skills. This is indicative of the growing role of employers and education to focus on developing human capital as well as competencies and digital skills because, without them, one cannot implement the digital transformation to gain economic, social, environmental, and consumer benefits [37,38] and to ensure sustainable economic development [39].

## 5. Conclusions

### 5.1. Contributions to Theory

From a theoretical point of view, this research adds a new baseline to the specialized literature on the possibilities of implementation of digital transformation, which is closely linked to the development of human capital competencies and digital skills with a focus on sustainability.

The research findings show that the concept of digital transformation (DT) is more comprehensive than the concepts of digitization and digitalization. DT includes three main elements: technological, where DT is based on the use of new digital technologies such as social media, mobile, analytics, or embedded devices; organizational, where DT requires a change of organizational processes or the creation of new business models; and *social*, where DT is influencing all aspects of human life. In order to ensure digital transformation, it requires maintaining the holistic approach to ensuring the implementation of all the above-mentioned elements.

*5.2. Contributions to Practice*

From a practical point of view, this is a specific research study that expands and provides insights into the situation in Latvia: employers can find in the results of this analysis support for the setting of strategies for the implementation of digital transformation and guidance for public authorities when defining regulatory policies for DT, as well as for educators to implement competence-based education together with digital education, thus integrating the elements of digital education in all study programs (courses) focused on developing the individual's key competences.

According to the DESI 2020, there is impressive progress in digital transformation across the EU Member States. Finland, Sweden, Denmark, and the Netherlands have the most advanced digital economies and lead the ranking of all the EU Member States. However, progress in the implementation of digital transformation in the EU and Latvia was very diverse. Some Member States were successful, while the others were not. In Latvia, too, digital solutions for enterprises and organizations were very different. Latvia is among the top EU Member States with the main public services reachable online for citizens and businesses. The digital transformation is being successfully carried out in the banking and insurance sector, as well as in several large enterprises. At the same time, in terms of the use of digital technologies, Latvia's SMEs had a lower score than the EU average. In addition, the problem is the development of human capital competencies and digital skills.

The analysis of the surveys of employers' ratings of the importance of digital transformation in Latvia allows us to conclude that the majority of the respondents surveyed rated the implementation of digital transformation in their organizations as medium-high, and in their organizations from three to five processes and more than five processes were digitalized; moreover, this was most specific to the public sector. This is a good trend, which shows that the digitalization process continues to progress, the majority of them are ready for change in the field of digitalization, and one can hope for positive changes in the future. However, about a third of enterprises are only at the early stage of digitalization, while some have not yet begun it, and the lowest ratings of the readiness of employees and enterprises for digitalization were found in the goods sector. This is particularly true of the private sector's small and medium-sized enterprises.

The research findings show that one of the most significant problems associated with digital transformation is human capital and the importance for employees to acquire relevant competencies and digital skills. Along with the development of specific digital skills, it is essential that individuals also build up soft skills such as the ability to communicate with others, work in a team and have tolerance, be able to start new activities, show initiative and enthusiasm, put forward real aims and try to achieve them, and act creatively, as well as other personal qualities. At the same time, according to a survey of residents of Latvia, businesspersons and entrepreneurs did not take sufficient care of the further education of their employees. This is indicative of the growing role of employers and education in focusing on developing human capital as well as competencies and digital skills because, without them, one cannot implement the digital transformation in all areas.

*5.3. Limitations*

Because in the last years in the specialized literature, much attention has been paid to explaining the concept of digital transformation, there are plenty of definitions provided by various authors, yet the present research is limited to and focuses on the general overview of the concepts and on an analysis of the implementation of digital transformation and the development of appropriate competences in employees. These limitations may be resolved through further research.

*5.4. Future Research Recommendations*

As regards the implementation of digital transformation in Latvia, in the future, it is advisable to conduct in-depth research on (1) the possibilities of implementing digital

transformation in the private sector's small and medium-sized enterprises, adapting to remote work and developing employers' competencies and digital skills; (2) how to provide the necessary infrastructure associated with digital transformation in order to be able to implement remote work in the long term; and (3) how investments made in the development of human capital competencies and digital skills facilitate digital transformation.

**Author Contributions:** Conceptualization V.B. and B.R.; methodology I.L.-E.; software P.R.; validation, P.R.; formal analysis P.R.; investigation B.R.; resources B.R.; data curation B.R.; writing—original draft preparation V.B. and B.R.; writing—review and editing, V.B. and B.R.; visualization I.L.-E.; supervision B.R.; project administration B.R.; funding acquisition B.R. All authors have read and agreed to the published version of the manuscript.

**Funding:** This research was funded by Latvia Ministry of Education and Science, State research programme INTERFRAME-LV, grant number VPP-IZM-2018/1-0005.

**Institutional Review Board Statement:** Not applicable.

**Informed Consent Statement:** Not applicable.

**Acknowledgments:** The research was supported by the National Research Programme "Latvian Heritage and Future Challenges for the Sustainability of the State", project "Challenges for the Latvian State and Society and the Solutions in International Context (INTERFRAME-LV)".

**Conflicts of Interest:** The authors declare no conflict of interest.

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
