# Peer review of "The Development of Digital Transformation and Relevant Competencies for Employees in the Context of the Impact of the COVID-19 Pandemic in Latvia"

_sustainability, doi:10.3390/su13169233_

Round 1
Reviewer 1 Report
Manuscript review: The Development of Digital Transformation and Relevant Competencies for Employees in the context of the impact of the COVID-19 pandemic in Latvia
I would like to start by congratulating the authors for choosing such interesting topic and the editor of Sustainability for the opportunity to review this paper.
The paper has potential, but there are a range of areas where improvements could be made before publication. My comments about improvements are noted below.
Summary:
The paper tackles an important topic regarding digital transformation and employees’ competencies in the context of COVID-19.
Comment:
In the following sections, comments and constructive arguments are presented that are organized in terms of organization and contributions.
Organization:
- The article lacks consistency, as an example see the case of “COVID-19” which is sometimes referred to as “Covid-19”. Both situations are fine, but the authors will have to choose one of the modalities.
- There are underlined sentences in the text and in the reference list when there is no need to do so (see e.g. lines 285-287, 290)
- Line 39 looks a citation (LIKTA, 2020), which I do not think it is. Maybe it is better to use “…Latvian Information and Communication Technology Association (LIKTA), which…”
- Line 98, [6] is in bold, it should not be.
- Again in lines 102, 105, 115, 124, etc. the authors have citations using (author, date) while it should be in sequential numbering.
The article seems rushed, counting the number of shortcomings, inconsistencies and typos found.
Contributions:
Regarding the research questions is a bit confusing as the authors use 1) and 2), but there are more than two questions. It should be clearer.
I think it might be useful to clarify the main limitations of the study. Furthermore, it may also be useful to divide the conclusions into: contributions to theory, contributions to practice, limitations, future research. I think that segmented by these topics the article will benefit from clearer and more objective conclusions.
The authors have made a significant effort to study such an opportune topic. My recommendation is to make a thorough revision and to resubmit the article for further review.
Author Response
Reviewer remarks:
Organization:
- The article lacks consistency, as an example see the case of “COVID-19” which is sometimes referred to as “Covid-19”. Both situations are fine, but the authors will have to choose one of the modalities.
In the new version of the article we are using COVID-19.
- Here are underlined sentences in the text and in the reference list when there is no need to do so (see e.g. lines 285-287, 290)
In the new version of the article are no underlined sentences in the text.
- Line 39 looks a citation (LIKTA, 2020), which I do not think it is. Maybe it is better to use “…Latvian Information and Communication Technology Association (LIKTA), which…”
Line 98, [6] is in bold, it should not be.
Again in lines 102, 105, 115, 124, etc. the authors have citations using (author, date) while it should be in sequential numbering.
The new version of the article takes into account all the reviewer's recommendations.
Contributions:
- Regarding the research questions is a bit confusing as the authors use 1) and 2), but there are more than two questions. It should be clearer.
The new version of the article has 3 questions.
- I think it might be useful to clarify the main limitations of the study. Furthermore, it may also be useful to divide the conclusions into: contributions to theory, contributions to practice, limitations, future research. I think that segmented by these topics the article will benefit from clearer and more objective conclusions.
In the new version of the article we divided the conclusions into: contributions to theory, contributions to practice, limitations, future research.
Reviewer 2 Report
Thank the authors for this really actual and interesting topic. I think the paper is well prepared, but there is some room for improvements:
- some text or numbers in the text are bold (e.g. on line 98, 100, 102)
- sometimes there are other types of citation - not only numbers of references, but names of authors in the text (e.g. Wieberneit (2021), Talin (2020))
- I think the sample is representative, but why is not anybody in the section "Other" in age 31-50 years?
- Discussion is missing - compare your results with another studies
- In Conclusion you should write down any limitations of your study
Round 2
Reviewer 1 Report
I do not think the authors carried out the revision seriously.
My arguments are supported by examples such as the revision of the citations. I can easily find is a mix of numbering [1] and (author, date). Please see: Duncin (2021), Talin (2020), Bloomberg (2018), etc.
Moreover, I recommended to divide the conclusions in sections, not the discussion...
The authors did not even bother to place the article according to the MDPI template. In other words, a rigorous revision is not verified.
Author Response
Dear Reviewer, we are very sorry for misunderstanding. We were so happy after your first positive remarks! Yes, we will do as you advice. We will divide the Conclusions in the sections and change the citations. We will try to put article according to the MDPI Template. Sorry.
Round 3
Reviewer 1 Report
I reread the article and it seems to me that it has improved since last reviews. However, I think the authors still don not follow the Journal's rules. For example, citations are not sequential, as the first citation is [1, 35] and must be [1, 2]. In addition to this shortcoming, there are many others, such as numbering: sometimes the authors use 66.7% (dot) and some other times 7,2% (comma). As I have already indicated rejection last time, and this is the 3rd time I review the article, I am going to recommend Major Revision and leave the supervision of the consistency in the academic editor's hands.
Author Response
Dear Reviewer, many thanks for your remarks. We look again throw the article and change citation as you advice. No comma, only dot.
We are very sorry.